# ST-Adapter: Parameter-Efficient Image-to-Video Transfer Learning

**Junting Pan**[1*]**, Ziyi Lin**[1*]**, Xiatian Zhu**[2]**, Jing Shao**[1]**, Hongsheng Li**[1,3]

[1]Multimedia Laboratory, The Chinese University of Hong Kong
[2]Surrey Institute for People-Centred Artificial Intelligence, CVSSP, University of Surrey
[3]Centre for Perceptual and Interactive Intelligence Limited

## Abstract

Capitalizing on large pre-trained models for various downstream tasks of interest have recently emerged with promising performance. Due to the ever-growing model size, the standard full fine-tuning based task adaptation strategy becomes prohibitively costly in terms of model training and storage. This has led to a new research direction in parameter-efficient transfer learning. However, existing attempts typically focus on downstream tasks from the same modality (*e.g.*, image understanding) of the pre-trained model. This creates a limit because in some specific modalities, (*e.g.*, video understanding) such a strong pre-trained model with sufficient knowledge is less or not available. In this work, we investigate such a novel cross-modality transfer learning setting, namely *parameter-efficient image-to-video transfer learning*. To solve this problem, we propose a new ***Spatio-Temporal Adapter*** (ST-Adapter) for parameter-efficient fine-tuning per video task. With a built-in spatio-temporal reasoning capability in a compact design, ST-Adapter enables a pre-trained image model without temporal knowledge to reason about dynamic video content at a small ($\sim$8%) per-task parameter cost, requiring approximately 20 times fewer updated parameters compared to previous work. Extensive experiments on video action recognition tasks show that our ST-Adapter can match or even outperform the strong full fine-tuning strategy and state-of-the-art video models, whilst enjoying the advantage of parameter efficiency. Code and model are available at `https://github.com/linziyi96/st-adapter`

## 1 Introduction

In the NLP field, almost all the state-of-arts across a wide range of downstream tasks have been achieved by adapting from large pretrained models (a.k.a. *foundation models* [7]) such as BERT [15] and GPT [54, 8]. The *de facto* standard approach to adapting a pretrained model to down-stream tasks is *fine-tuning* either *fully* or *partially* (*e.g.*, linear probing by training the newly added multi-layer perceptron layers on the top alone), subject to the condition of adopting a similar network architecture as the pretrained model. Nonetheless, given increasingly larger whilst ever stronger foundation models (*e.g.*, GPT-3 with 175B parameters), fully fine-tuning the whole model for every single downstream task would become prohibitively expensive and infeasible in terms of training cost and model storage. This could significantly restrict their deployment and usability in real-world applications. In this context, a series of NLP works has been introduced towards efficient transfer learning with better trade-offs between parameter and accuracy [25, 24, 39, 36].

This trend has recently motivated the computer vision community. For example, the CLIP model [55], trained with 400 million web image-text pairs, achieves promising performances on a variety of

---

[*]Equal contribution

36th Conference on Neural Information Processing Systems (NeurIPS 2022).

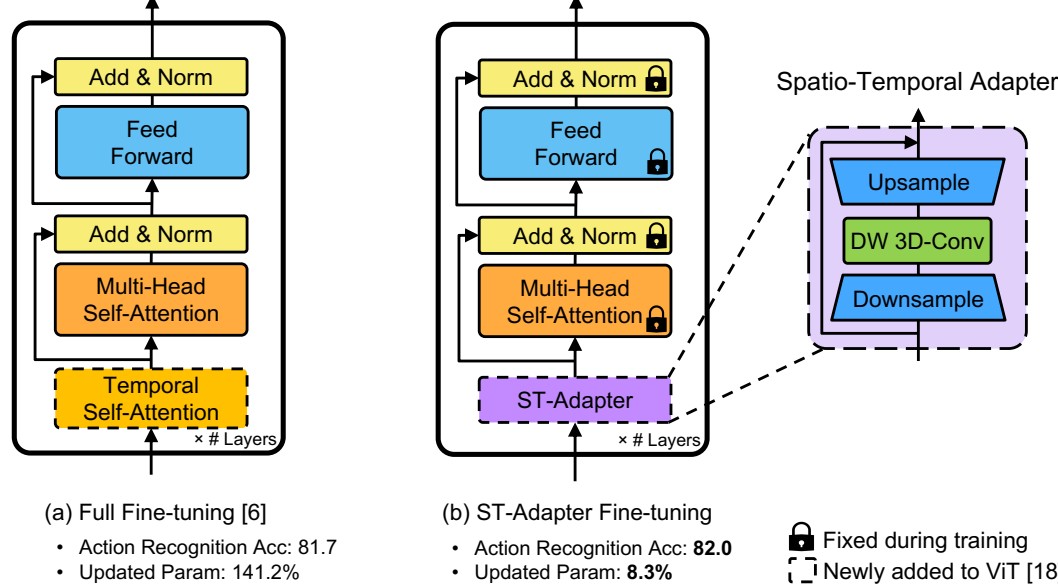

Figure 1: **Image-to-video transfer learning strategies.** (a) The state-of-the-art methods for adapting a pre-trained image model (*e.g.*, ViT [16] in this example) to video tasks (*e.g.*, action recognition) usually adopt the paradigm of first designing a temporal learning module and then fine-tuning the whole network fully [2, 6, 9]. This is parameter-inefficient since a specific instance of such a large model is resulted for each downstream task. In contrast, (b) we propose to only train a lightweight *Spatio-Temporal Adapter* with much fewer parameters for each individual downstream task at a significantly smaller computational cost. Surprisingly, our method can match or even surpasses the full fine-tuning based methods (including prior art video models in terms of accuracy), whist enjoying higher parameter efficiency and cheaper training cost.

image recognition and generation tasks. In the video domain, with significantly more computational cost and resources, Xu *et al.* [79] trained a video variant of CLIP but excelled on a smaller number of downstream video tasks. This is partly attributed to two orders of magnitude more minor training data and limited availability of computing resources, as large video data is notoriously more difficult to collect, manage, and process than image data. Under these restrictions, large pre-trained image models are arguably still favorable in the selection of model initialization for video tasks.

In this work, we investigate a novel, critical problem of ***efficiently adapting large pre-trained image models for video downstream tasks***, with a focus on the widely influential action recognition task. Considering that training video models is drastically more expensive in both computing resource and time than image models [19], this problem becomes particularly more useful and valuable in practice. On the other hand, it is also more challenging and non-trivial due to the extra necessity of overcoming the big gap between image and video in transfer learning. Especially, pre-trained image models lack the ability to infer temporal structured information, which however is critical in video understanding. In fact, the key design with state-of-the-art video models [10, 41, 6, 9] is usually about learning the temporal dimension based on contemporary image models. Although model initialization is still important, they largely go beyond the fine-tuning strategy, as *architectural modification* is often imposed in addition to full model training/fine-tuning per downstream task.

Given that this is a new problem, we first conduct a comprehensive benchmark using both various fine-tuning methods for image-to-video transfer learning and state-of-the-art video models [6, 9]. Regarding the pretrained image model, we select two Vision Transformer (ViT) [16] models, with one from CLIP pre-training [55] and the other pre-trained on ImageNet-21K [14]. ViT is representative in terms of network architecture, pre-training algorithm, and training data scale. Crucially, we further propose an efficient yet effective ***Space-Time Adapter*** (ST-Adapter), capable of extracting and leveraging the pre-trained knowledge of a large image model to achieve superior video understanding at a small parameter cost. Specifically, ST-Adapter is formulated based on a novel parameter-efficient bottleneck with a sequence of operations including feature dimension reduction, spatial-temporal modeling, and feature dimension recovery. It is easy to implement and scalable for deployment since

all the primitive steps are realized with standard operators (*e.g.*, fully-connected layer, depth-wise 3D convolution). With such a lightweight design, our bottleneck can be cheaply integrated throughout the base network for enabling stronger layer-wise spatio-temporal learning. As a result, our model can be more rapidly optimized using fewer training epochs for significant convergence advantage.

We summarize the **contributions** as follows. **(1)** We investigate a new problem of *parameter-efficient image-to-video transfer learning*. Our motivation is to advocate the usability and deployment of increasingly larger whilst ever more powerful *pre-trained* image models in benefiting more challenging video understanding tasks. **(2)** We establish a benchmark for action recognition tasks by comprehensively experimenting with a variety of fine-tuning strategies and several state-of-the-art video understanding models. **(3)** We introduce a novel parameter-efficient *Spatio-Temporal Adapter* (ST-Adapter) for more effectively capitalizing a large pre-trained image model in video understanding. By grounding all the primitives on standard operators, ST-Adaptor is easy to implement and friendly to deployment. **(4)** Extensive experiments on action recognition datasets show that our ST-Adapter outperforms not only existing parameter-efficient alternatives and the full fine-tuning strategy, but also state-of-the-art video methods with the same network architecture and model initialization.

## 2    Related Work

**Parameter-efficient transfer learning**   Driven by the wider application of large pre-trained language models across a diversity of downstream tasks, the topic of efficient tuning has received increasing attention in NLP. Existing efficient tuning methods fall broadly into three categories. The *first* category is to introduce task-specific adapters [25, 24, 51, 50]. Specifically, an adapter consists of lightweight modules inserted between layers of a pre-trained model. To be parameter-efficient, only those newly added adapter modules need to be updated during task fine-tuning, whilst all the parameters of the large pre-trained model, which takes the majority proportion of the whole solution, are frozen. The *second* category is prompt tuning [39, 52, 31, 62, 42]. Instead of manipulating the network architecture, these methods prepend a set of learnable tokens at the input point of the model or intermediate layers. Similarly, only these added tokens need to be optimized for each downstream task. The *third* category is learning weight approximation [27]. In particular, only the low-rank matrices for approximating the weights need to be updated during training.

Early works for efficient transfer learning in vision focus on parameter sharing in the context of multitask learning [83, 57, 56]. Recently, there are several works for extending the efficient tuning idea from NLP to vision tasks. CoOp [85] and CoCoOp [86] apply prefix tuning for adapting the CLIP model to various image recognition tasks. VL-Adapter [65] achieves the performance comparable to full fine-tuning on challenging vision-language tasks. Commonly, their design focuses are all restricted to the text encoder of the CLIP model. More recently, [29, 4, 84] introduce the idea of prompt learning to visual backbones. They obtained favorable results on various image recognition benchmarks. Moving a step further, in this work, we consider the more challenging adaptation problem from a pre-trained image model without temporal knowledge to video understanding tasks.

**Video action recognition**   Action recognition in the unconstrained video has largely been dominated by deep learning methods, thanks to the availability of large video datasets, *e.g.*, Kinetics [10–12] and Something-Something [22]. As a key component, the model architectures adopted by existing video methods has expanded from CNNs [32, 68, 19, 18, 77, 69, 72, 48, 41, 45] to Transformers [17, 40, 38, 44, 2, 6]. As temporal information is important for modeling the dynamics, a variety of motion learning techniques has been introduced [75, 30, 49]. Further, different training methods have also been explored, *e.g.*, unsupervised learning [67, 20, 76], and video-text contrastive learning [64, 79, 78, 66]. New opportunities for stronger video models are created following the introduction of large pretrained foundation models [55, 28, 81]. For example, Wang *et al.* [74] equipped the CLIP with temporal modules and good performance can be achieved after the model is fully fine-tuned on video datasets. Ju *et al.* [31] adopted the CLIP model for video recognition tasks by learning video-specific prompts. In contrast, in this work, we explore the potential of the large pre-trained image models with the parameter-efficient adapter strategy. Importantly, despite the simplicity, we bring about more significant advantages in performance along with a new benchmark on parameter-efficient image-to-video transfer learning.

# 3  Methodology

To capitalize a large pre-trained image model for more challenging video understanding such as action recognition in a cross-modality manner, it is necessary to fill the intrinsic gap between image and video. For easier understanding, we start with an intuitive baseline based on temporal aggregation.

**Temporal aggregation**  A straightforward baseline method of exploiting a pre-trained image model for video understanding is to temporally aggregate per-frame feature representations (*e.g.*, average pooling). Concretely, given an input video clip $\mathbf{V} \in \mathbb{R}^{T \times H \times W}$, where $T, H, W$ are the number of frames, height and width respectively. Following [16], we first split each frame into $N = H \times W/P^2$ patches of size $P \times P$. Then, we flatten these patches and project them into a sequence of patch tokens $\mathbf{Z}_t = [\mathbf{z}_1, ...\mathbf{z}_s, ..., \mathbf{z}_N], \mathbf{z}_s \in \mathbb{R}^d$ where $d = 3 \times P^2$ with $t = 1, ..., T$. The sequence of feature vectors is then enhanced with the positional embedding by element-wise addition, along with a trainable class token concatenated. Subsequently, we feed each sequence with $N + 1$ tokens to a stack of self-attention based blocks individually. For each sequence we keep only the classification token $\mathbf{z}_t^{cls}$. We further perform temporal average pooling on the class tokens $\mathbf{z}_{final} = \frac{1}{T} \sum_t \mathbf{z}_t^{cls}$ to yield a compact representation for the whole clip. We obtain the prediction by passing $\mathbf{z}_{final}$ through a classifier. As the sptial information is only naively averaged over time, it is also known as `Space-Only TimeSformer` [6].

**Spatio-temporal attention**  For more dedicated structural modeling in the time dimension with ViTs, a mainstream approach in the video domain is to develop various spatio-temporal attention mechanisms by further imposing temporal attention on top [6, 2, 3, 9, 82, 23]. We choose two representative video ViT models, TimeSformer [6] and XViT [9], in our performance benchmark. However, state-of-the-art video ViT models often need to *fully fine-tuned* per task, which is ***parameter-inefficient***, given that in this way we have to keep a separate copy of the whole fine-tuned model parameters for every single task.

## 3.1  Preliminaries

Our method is inspired by the Adapter [25] designed for parameter-efficient transfer learning in NLP. Specifically, the adapter module is composed of a down-projection linear layer followed by a non-linear activation function and an up-projection linear layer. Formally, given an input feature matrix $\mathbf{X} \in \mathbb{R}^{N \times d}$ at the $i$-th layer, the feature adaptation process can be written as:

$$\texttt{Adapter}(\mathbf{X}) = \mathbf{X} + f(\mathbf{X}\mathbf{W}_{down})\mathbf{W}_{up}, \tag{1}$$

where $\mathbf{W}_{down} \in \mathbb{R}^{d \times r}$ refers to the down projection layer, $\mathbf{W}_{up} \in \mathbb{R}^{r \times d}$ the up-projection layer, and $f(\cdot)$ the activation function. Note, that a residual summation is applied for preserving the information in input as required. The idea of Adapter has been remarkably successful in NLP due to several advantages: (1) High parameter efficiency across tasks since only a small number of parameters are task-specific; (2) Reaching on-par performance compared to full fine-tuning; (3) Taking significantly small training costs; (4) Avoiding the catastrophic forgetting limitation of full fine-tuning.

We aim to propagate the success of Adapter from NLP to computer vision particularly the image-to-video transfer learning problem as discussed earlier. To that end, we introduce a novel Adapter tailored specially for spatio-temporal reasoning – a key capability for video understanding which, however, existing NLP Adapter variants lack.

## 3.2  Spatio-Temporal Adapter (ST-Adapter)

Typically, an image model only considers the ability of spatial modeling. The *objective* of our Spatio-Temporal Adapter (ST-Adapter) is to enable a pre-trained image model to reason about spatial and temporal information of video in a parameter efficient principle. In design, we consider a couple of practically-crucial criteria: (1) *Smaller parameter size*: The parameter cost for each downstream task should be small – the essential criterion for parameter efficiency. (2) *Development friendliness*: This is critical for real-world development and deployment. In practice, it is necessary that a model can be easily implemented using the standard highly optimized deep learning toolboxes (*e.g.*, PyTorch, TensorFlow, TensorRT, and TorchScript), without tedious per-toolbox specialization.

This also facilitates the realization of high inference efficiency across a diversity of running platforms due to the best usage of built-in software and hardware resources.

Under these considerations, we formulate the proposed ST-Adapter by sticking to *commonly-adopted primitive operators alone*. Starting with the above Adapter (Eq. (1)) originally developed for NLP tasks, we further introduce a spatio-temporal operator realized by a standard depth-wise 3D-convolution layer [18] between the bottlenecks (Figure 1). In particular, our spatio-temporal operator enables layer-wise temporal inference efficiently, because it only operates in a compressed low-dimensional (*e.g.*, 128D) feature space and the depth-wise convolution is highly efficient both in parameter and computation [26]. As a result, this yields an introduction of tiny extra ($\sim$2%) parameters and ($\sim$0.3%) computation. Formally, our ST-Adapter can be expressed as:

$$\texttt{ST-Adapter}(\mathbf{X}) = \mathbf{X} + f\Big(\texttt{DWConv3D}(\mathbf{X}\mathbf{W}_{down})\Big)\mathbf{W}_{up}, \qquad (2)$$

where $\texttt{DWConv3D}$ denotes the depth-wise 3D-convolution for spatio-temporal reasoning we introduce. It is noteworthy that before applying $\texttt{DWConv3D}$, the down-projected feature representations will be first reshaped from $\mathbf{X}' \in \mathbb{R}^{T \times N \times d}$ to $\mathbf{X}'' \in \mathbb{R}^{T \times h \times w \times d}$ (where $N = h \times w$) to have the spatial and temporal dimensions prepared for reasoning. With this highly integrated design, our ST-Adapter enjoys the same efficiency and flexibility as the NLP Adapter, while uniquely being able to conduct spatio-temporal modeling. l.

### 3.3 ST-Adapter Integration

For proper adaptation, the adapter modules are often integrated between layers of a Transformer. In NLP, a variety of integrating designs have been investigated. For example, [25] deploys two adapter modules per layer with one following the Multi-Head Self-Attention (MHSA) and the other following the Feed-Forward Networks (FFN) [25]. On the other hand, [63, 5] suggest that adding only one adapter after the FNN suffices. Similarly, our ST-Adapter can be also integrated generally at distinctive positions. Empirically, we find that a decent performance can be achieved in case a single ST-Adapter is placed before the MHSA of each transformer block (Figure 1(a) and Table 5c).

## 4 Experiments

### 4.1 Experiments Setup

**Datasets** For the benchmark experiments, we use two popular video action recognition datasets.

*Kinetics-400 (K400)*: The K400 [33] dataset contains $\sim$240k training videos and 20k validation videos labeled with 400 action categories. Most videos have a length of 10s or about 300 frames. While there is a great diversity in these videos, they are largely biased to spatial appearance [60].

*Something-Something-v2 (SSv2)*: The SSv2 [22] dataset consists of 220,487 videos covering 174 human actions. The video length ranges from 2 to 6 seconds. In contrast to K400, SSv2 presents richer temporal information with much higher significance [60].

*Epic-Kitchens-100 (EK100)*: The EK100 [13] dataset consists of 100 hours of video in egocentric perspective recording a person interacting with a variety of objects in the kitchen. Each video sample is labeled with a verb and a noun. We report top-1 verb and noun classification accuracy.

**Pre-trained models** In all experiments, we use the standard ViT [16] as our base backbone model. We conduct most of our experiments with the *ViT-B/16* variant with 12 layers and 86M parameters, taking as input a sequence of patches at size $16 \times 16$.

What was learned during pre-training directly decides the knowledge that can be transferred to downstream tasks, thus also the effectiveness upper bound of transfer learning methods. To this end, we benchmark the same backbone under two different pre-training strategies: pre-training with web-scale raw data that has been recently proposed by CLIP [55] (400M image-text pair) and classical supervised pre-training on annotated data from ImageNet-21K (21k classes and 14M images).

**Implementation details.** All details, including training and testing settings and module instantiation details, are provided in the appendix.

**Competitors**  We provide several transfer learning approaches in our benchmark for efficient image-to-video transfer learning. Note that the parameters of the linear classifier are always updated during training for all approaches.

(1) *Full Fine-tuning*: Fully updating all the parameters when adapting for a specific target task.

(2) *Partial Fine-tuning*: Only update the last ViT layer while keeping the rest of the parameter fixed.

(3) *Temporal Fine-tuning*: We only tune the temporal attention modules (*i.e.,* TA) in the SA+TA architecture.

(4) *Linear Probing*: Freezing all the parameters except those in the linear classification layer.

(5) *Adapter* [25]: Adding small sub-networks between layers of a pre-trained model. During fine-tuning, we only update the newly added parameters introduced by the adapters.

(6) *Prompt Tuning* [29]: Prepending a sequence of learnable prompt tokens to the input visual patch tokens. During fine-tuning, only these newly added prompts are updated.

(7) *Attention Pooling Head*: Replacing the original temporal average pooling with a temporal attention pooling layer (similar to the one used in [9]) before the classification head.

These approaches above do not incorporate temporal modeling to the image ViT. Hence, we further consider temporally augmented ViT architectures as introduced in state-of-the-art video methods:

(a) *Spatial Attention Only (SA)*: Space-Only TimeSformer [6].

(b) *Spatial Attention + Temporal Attention (SA+TA)*: The default TimeSformer [6] with divided space-time attention (Fig. 1a).

(c) *Spatial Attention + Temporal Shift (SA+TS)*: XViT [9].

Note that not all fine-tuning protocols are compatible with each of these video ViT variants. Take SA+TS for example, the original model behavior is altered with channel shift, as a result, it is not compatible with *Linear Probing* that requires freezing all the parameters of the backbone.

## 4.2   Main Results and Analysis

**Cross-modality fine-tuning benchmark.**  Table 1 presents the results of fine-tuning a ViT-B/16 pre-trained with CLIP and ImageNet-21K. All baselines are built by combining existing efficient fine-tuning methods with three state-of-the-art ViT-based action recognition models. From the results we can see that:

(i) For CLIP pre-trained model, ST-Adapter performs on par with Full Fine-tuning (**82.0** vs. 81.7 for K400 and **66.3** vs. 66.1 for SSv2) while updating far less parameters (**7.2M** vs. 121.57M). ST-Adapter significantly outperforms all other efficient fine-tuning methods. We see that baselines like Prompt Tuning and Partial Fine-tuning can provide non-trivial gain in performance compared to Linear Probe, but are still behind our ST-Adapter.

(ii) Our ST-Adapter can generalize across different pre-training datasets and methods. We can see that CLIP pre-train models dominate over ImageNet-21K pre-train ones. These results well match the shift of paradigm in current AI research [7], where pre-training no longer needs limiting to curated data and annotations to deliver good performance on downstream tasks, but can take advantage of broader scale web raw data.

Interestingly, we observe that SSv2, a motion-centric dataset in design, also benefits from stronger appearance (image) pre-training. We think this may attribute to that raw textual description can provide a much richer description (*i.e.*, human-object relations) of the image than curated limited categorical labels. Full fine-tuning on SA+TS (XViT) performs slightly worse with CLIP pretrain than ImageNet-21k pretrain. We conjecture this is because the channel shift operation breaks the knowledge in the pre-training weights, and thus does not benefit much from stronger pre-training like CLIP.

**Comparison to the state-of-the-art models.**  We compare ViT with ST-Adapter to other state-of-the-arts methods on both K400 dataset [33] in Table 2, SSv2 dataset [22] in Table 3 and EK100 dataset [13] in Table 4. We can observe that:

(i) With the proper adaptation method, we can simply turn a large image foundation model into a good video model by only tuning a few parameters. Our results are comparable to or better than previous

Table 1: **Benchmark results on Kinetics-400 and Something-Something-v2.** We evaluate all the approaches on two datasets with ViT-B/16 pretrained with CLIP and ImageNet-21K. For each entry, we report the top1 action recognition accuracy and the number of fine-tuned parameters. All methods introduce extra parameters beside parameters of the ViT backbone and linear classifier. Our ST-Adapter achieves the best trade-off between accuracy and training efficiency. It is the only efficient fine-tuning method that can match the performance of full fine-tuning. The *TM?* column shows whether the method includes temporal modelling, *i.e.*, a temporal aggregation method other than average pooling. All models are trained using 8 frames and tested with 3 views.

| Fine-tuning Methods | Architecture | TM? | Fine-tuned Params (M) | CLIP | | ImageNet-21K | |
| --- | --- | --- | --- | --- | --- | --- | --- |
| | | | | K400 | SSv2 | K400 | SSv2 |
| Full Fine-tuning | SA | | 86.11 | 81.0 | 44.0 | 76.9 | 40.0 |
| | SA + TA [6] | ✓ | 121.57 | 81.7 | 66.1 | 78.0 | 59.5 |
| | SA + TS [9] | ✓ | 93.79 | 78.0 | 62.0 | **78.5** | **64.4** |
| Partial Fine-tuning | SA | | 7.40 | 80.1 | 37.6 | 61.7 | 20.4 |
| | SA + TA | ✓ | 10.36 | 80.3 | 57.5 | 63.1 | 29.3 |
| Temporal Fine-tuning | SA + TA | ✓ | 35.8 | 81.3 | 59.4 | 76.5 | 51.9 |
| Prompt Tuning | SA | | 1.18 | 79.3 | 39.3 | 71.4 | 26.3 |
| Attentional Pooling | SA | ✓ | 2.36 | 75.3 | 21.5 | 59.1 | 15.1 |
| Linear Probe | SA | | 0.31 | 76.6 | 21.9 | 60.1 | 14.8 |
| Adapter [25] | SA | | 6.77 | 81.6 | 46.2 | 76.2 | 40.5 |
| ST-Adapter (ours) | SA | ✓ | 7.20 | **82.0** | **66.3** | 76.6 | 62.8 |

methods tailored for such tasks. Our largest model with ViT-L backbone set a new state-of-the-art in K400 by achieving 86.7% top-1 accuracy.

(ii) It is noteworthy that, our method takes significantly fewer frames as input compared to other methods (8 vs. 16, 32, 64, 96). It is also reflected in terms of GFlops. Saying that the ViT was not designed for efficiency purposes like [38, 9, 43, 17] but the adapted CLIP ViT has achieved similar accuracy-efficiency trade-offs.

(iii) The paradigm of pre-training and fine-tuning has been widely adopted in most state-of-art methods to achieve good performance. Between them, most of the approaches start from image pre-trained models, and only a few can afford video pre-training. Note that for the Something-Something dataset, except MViT [17] pre-trained on video data from scratch, the rest of methods are still initialized from image pre-trained weights. A good image pre-trained model with rich appearance information can facilitate temporal modeling in temporally challenging datasets like SSv2.

(iii) It is evident in Table 4 that our ST-Adapter consistently brings a big margin on egocentric videos. Also, we found that without our ST-Adapter, it is much more difficult to directly adapt CLIP pre-trained ViT on the domain of egocentric video with high sensitivity to the hyper-parameter setting. ST-Adapter eases the training process. It is worthy to note that, all current transformer based approaches need to be pre-trained first on image dataset and then fine-tuned on Kinetics dataset before fine-tuned with egocentric videos. In contrast, our ST-Adapter can be directly applied to an image model and trained with target egocentric video alone.

### 4.3 Ablations

Unless otherwise specified, we use ViT-B/16 backbone and 8 input frames in all ablation experiments, and we use one ST-Adapter with bottleneck width 384 before MHSA in each Transformer block.

**Where to insert ST-Adapter** By default, we insert a ST-Adapter to every Transformer block in the backbone, but we also show the performance impact of using fewer ST-Adapters. As shown in Table 5b, while more ST-Adapters tend to do better, ST-Adapters at deeper layers boost performance more

Table 2: **Results on Kinetics-400 validation set**. "Frames" denotes the total number of frames used during inference which is: # frames per clip × # temporal clip × # spatial crop. "GFlops" means $10^9$ Flops. Our ViT w/ ST-Adapter achieves new state-of-the-art performances on K400 at similar GFlops.

| Model | Pretrain | #Frames | GFlops | Top-1 | Top-5 |
|---|---|---|---|---|---|
| *Methods with full-finetuning* | | | | | |
| LGD[53] | IN-1K | 128×N/A | N/A | 79.4 | 94.4 |
| SlowFast+NL[19] | - | 16×3×10 | 7020 | 79.8 | 93.9 |
| ip-CSN[70] | Sports1M | 32×3×10 | 3270 | 79.2 | 93.8 |
| CorrNet[71] | Sports1M | 32×3×10 | 6720 | 81.0 | - |
| X3D-XL[18] | - | 16×3×10 | 1452 | 79.1 | 93.9 |
| MoViNet-A6[34] | - | 120×1×1 | 386 | 81.5 | 95.3 |
| ViT-B-VTN [47] | IN21K | 250×1×1 | 3992 | 78.6 | 93.7 |
| TimeSformer-L[6] | IN21K | 96×3×1 | 7140 | 80.7 | 94.7 |
| STAM [61] | IN21K | 64×1×1 | 1040 | 79.2 | - |
| X-ViT[9] | IN21K | 16×3×1 | 850 | 80.2 | 94.7 |
| Mformer-HR[49] | IN-21K | 16×3×10 | 28764 | 81.1 | 95.2 |
| MViT-B,32×3[17] | - | 32×1×5 | 850 | 80.2 | 94.4 |
| ViViT-L[2] | JFT300M | 16×3×4 | 17352 | 82.8 | 95.3 |
| Swin-B[44] | IN1K | 32×3×4 | 3384 | 80.6 | 94.6 |
| Swin-L(384)[44] | IN21K | 32×5×10 | 105350 | 84.9 | 96.7 |
| UniFormer-B[38] | IN1K | 32×1×4 | 1036 | 82.9 | 95.4 |
| VATT-Large(320)[1] | HowTo100M | 32×3×4 | 29800 | 82.1 | 95.5 |
| TokenLearner[58] | JFT300M | 64×3×4 | 48912 | 85.4 | 96.3 |
| OMNIVORE(Swin-L)[21] | IN22K+SUN | 32×3×4 | 7248 | 84.1 | 96.3 |
| MTV-H[80] | WTS-280 | 32×3×4 | 73570 | **89.9** | **98.3** |
| ViT-B w/o ST-Adapter | CLIP | 8×3×1 | 419 | 81.0 | 95.5 |
| ViT-L w/o ST-Adapter | CLIP | 8×3×1 | 1941 | 85.8 | 97.2 |
| *Methods with frozen backbone* | | | | | |
| Our ViT-B w/ ST-Adapter | CLIP | 8×3×1 | 455 | 82.0 | 95.7 |
| Our ViT-B w/ ST-Adapter | CLIP | 16×3×1 | 911 | 82.5 | 96.0 |
| Our ViT-B w/ ST-Adapter | CLIP | 32×3×1 | 1821 | 82.7 | 96.2 |
| Our ViT-L w/ ST-Adapter | CLIP | 8×3×1 | 2062 | **86.7** | **97.5** |
| Our ViT-L w/ ST-Adapter | CLIP | 16×3×1 | 4124 | 86.9 | 97.6 |
| Our ViT-L w/ ST-Adapter | CLIP | 32×3×1 | 8248 | 87.2 | 97.6 |

than those at shallower layers. This observation is useful when we insert ST-Adapters into deeper models and having an Adapter for each block might be too expensive. We also show the performance when inserting ST-Adapters to different positions within a block. As shown in Table 5c, while the performance is relatively insensitive to the position of the Adapters, using multiple adapters in one block may substantially boost performance on some datasets, like SSv2 in our case.

**Training parameter efficiency** We experiment with a different number of channels in the middle of the bottleneck design. As shown in Table 5a and Fig. 2a, our method is effective with a wide range of bottleneck width: even with a channel reduction to 64, our ST-Adapters still obtain relatively good performance, outperforming all baselines in Table 1 except for Full Fine-tuning (SA + TA). Even with a bottleneck width of 768, our ST-Adapters are still very parameter efficient, introducing only about 1/6 new parameters to a Transformer encoder block. In contrast to the *inverted bottleneck* design commonly used with depthwise convolutions [59], ST-Adapters work best with regular bottlenecks. The success of transfer learning with such low-rank projections again shows the rich knowledge and strong potential of modern foundation models.

**Training time efficiency** In Fig. 2b we show an enlarged difference between full fine-tuned models and our ST-Adapters with low training budgets. When we reduce the number of training steps, the accuracy of full fine-tuned models drops significantly faster than models with ST-Adapters. This shows the advantage of our proposed modules when backbone models are large or computational resources are limited. We also report the total training GPU-hours and peak memory usage for three models: TimeSformer, ViT-B/16, ViT-B/16 with ST-Adapter (8 input frames, 16 samples per GPU on 8 V100 GPUs) in Table 6.

Table 3: **Results on Something-Something-v2 validation set.** "Frames" denotes the total number of frames used during inference which is: # frames per clip $\times$ # temporal clip $\times$ # spatial crop. "GFlops" means $10^9$ Flops. Our ViT w/ ST-Adapter outperforms most of the current methods by only fine-tuning a very small set of parameters. Here the ViT-B w/ ST-Adapter result is reported using **2 ST-Adapters per block**.

| Model | Pretrain | #Frames | GFlops | Top-1 | Top-5 |
|---|---|---|---|---|---|
| *Methods with full-finetuning* | | | | | |
| TSM[41] | IN1K | 16×1×1 | 66 | 63.3 | 88.5 |
| GST[46] | IN1K | 16×1×1 | 59 | 62.6 | 87.9 |
| MSNet[35] | IN1K | 16×1×1 | 101 | 64.7 | 89.4 |
| CT-Net[37] | IN1K | 16×1×1 | 75 | 64.5 | 89.3 |
| TDN[73] | IN1K | 16×1×1 | 72 | 65.3 | 89.5 |
| TimeSformer-HR[6] | IN21K | 16×3×1 | 5109 | 62.5 | - |
| X-ViT[9] | IN21K | 32×3×1 | 1270 | 65.4 | 90.7 |
| Mformer-L[49] | IN21K+K400 | 32×3×1 | 3555 | 68.1 | 91.2 |
| ViViT-L[2] | IN21K+K400 | 16×3×4 | 11892 | 65.4 | 89.8 |
| MViT-B-24,32×3[17] | K600 | 32×1×3 | 708 | 68.7 | 91.5 |
| Swin-B[44] | IN21K+K400 | 32×3×1 | 963 | 69.6 | 92.7 |
| UniFormer-B[38] | IN1K+K600 | 32×3×1 | 777 | 71.2 | 92.8 |
| OMNIVORE (Swin-B)[21] | IN22K+K400+SUN | 32×3×1 | 963 | 71.4 | 93.5 |
| MTV-B(320p)[80] | IN21K+K400 | 32×3×4 | 11160 | 68.5 | 90.4 |
| ViT-B w/o ST-Adapter | CLIP | 8×3×1 | 419 | 44.0 | 77.0 |
| ViT-L w/o ST-Adapter | CLIP | 8×3×1 | 1941 | 48.7 | 77.5 |
| *Methods with frozen backbone* | | | | | |
| Our ViT-B w/ ST-Adapter | CLIP | 8×3×1 | 489 | 67.1 | 91.2 |
| Our ViT-B w/ ST-Adapter | CLIP | 16×3×1 | 977 | 69.3 | 92.3 |
| Our ViT-B w/ ST-Adapter | CLIP | 32×3×1 | 1955 | 69.5 | 92.6 |
| Our ViT-L w/ ST-Adapter | CLIP | 8×3×1 | 2062 | **70.0** | **92.3** |
| Our ViT-L w/ ST-Adapter | CLIP | 16×3×1 | 4124 | 71.9 | 93.4 |
| Our ViT-L w/ ST-Adapter | CLIP | 32×3×1 | 8248 | **72.3** | **93.9** |

Table 4: **Results on Epic-Kitchens-100 validation set.** "Frames" denotes the total number of frames used during inference which is: # frames per clip $\times$ # temporal clip $\times$ # spatial crop.

| Model | Pre-train data | #Frames | Verb | Noun |
|---|---|---|---|---|
| *Methods with full-finetuning* | | | | |
| ViViT-L [2] | IN21K+K400 | 16 × 3 × 10 | 66.4 | **56.8** |
| MFormer-B [49] | IN21K+K400 | 16 × 3 × 10 | 66.7 | 56.5 |
| XViT(8x) [9] | IN21K+K400 | 8 × 3 × 1 | 66.7 | 53.3 |
| ViT-B/16 w/o ST-Adapter | CLIP | 8 × 3 × 1 | 54.8 | 50.4 |
| *Methods with frozen backbone* | | | | |
| Our ViT-B/16 w/ ST-Adapter | CLIP | 8 × 3 × 1 | **67.6** | 55.0 |

Table 5: **Ablation study on K-400 and SSv2.** (a) We show the performance with different channel numbers in the bottleneck. (b) We evenly divide the 12 blocks of ViT-B/16 into 3 groups. Block no. 1 is closest to input and no. 12 is closest to output. (c) Effect of where to put the ST-Adapter inside a block, whose diagram is shown in Fig. 1.

(a) **Bottleneck width**

| width | K400 | SSv2 |
|---|---|---|
| 64 | 81.4 | 64.4 |
| 128 | 81.6 | 64.9 |
| 256 | 81.8 | 65.5 |
| 384 | **82.0** | **65.6** |
| 768 | 81.9 | 65.5 |

(b) **Global position**

| 1-4 | 5-8 | 9-12 | K400 | SSv2 |
|---|---|---|---|---|
| ✓ | | | 77.7 | 45.9 |
| | ✓ | | 80.0 | 60.9 |
| | | ✓ | 81.3 | 62.8 |
| | ✓ | ✓ | 81.8 | **65.6** |
| ✓ | ✓ | ✓ | **82.0** | 65.6 |

(c) **Local position**

| position | K400 | SSv2 |
|---|---|---|
| before MHSA | 82.0 | 65.6 |
| after MHSA | 81.9 | 65.7 |
| after FFN | 81.9 | 65.9 |
| before & after MHSA | 82.0 | **67.0** |

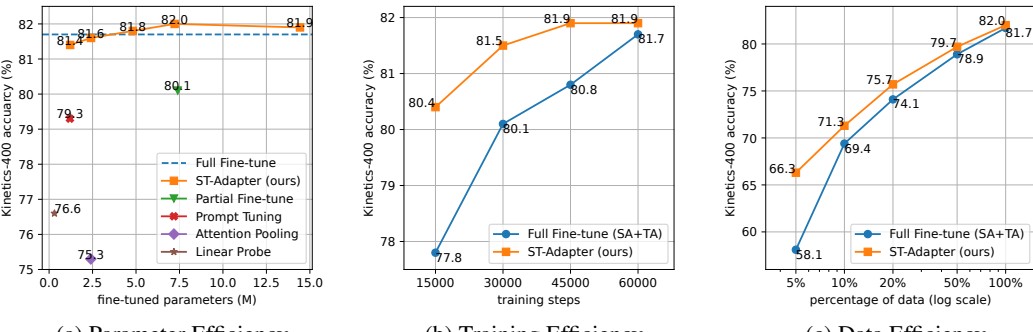

|  (a) Parameter Efficiency | (b) Training Efficiency | (c) Data Efficiency |

Figure 2: **Ablation study on efficiency** (a) Parameter efficiency: ST-Adapter (with different bottleneck width) is compared with efficient fine-tuning methods in Table 1. (b) Training efficiency: We compare ST-Adapter with Full fine-tuning under different training schedules. Batch size is aligned and their original schedules are shortened proportionally. (c) Data efficiency: Performance comparison on different training data scales. The same ViT-B/16 with CLIP pre-training is used for all experiments.

**Training data efficiency** Fig. 2c showcases the impact of training data size on action recognition accuracy. Even with the same pre-trained weights, ST-Adapters tend to obtain higher performance than full fine-tuning especially on smaller datasets: the margin between the two models increases with the shrinkage of data. This shows that ST-Adapters are powerful tools to transfer to downstream tasks where only a small amount of labeled data is available.

**Effects of kernel shape** We ablate the effect of kernel size in the depth-wise convolutions inside our proposed ST-Adapter. It is shown in Table 7 that the temporal span is most sensitive, suggesting the significance of temporal structural modeling as we focus on in this work.

Table 6: **Training time and memory.** For full-finetuning we used the recipes in [6].

| Model | Training GPU-hours (K400) | Peak mem (MB) |
|---|---|---|
| TimeSformer[6] (Full Fine-tune) | 60 (+161%) | 21694 (+52%) |
| ViT-B/16 (Full Fine-tune) | 40 (+74%) | 17275 (+21%) |
| ViT-B/16 w/ ST-Adapter | **23** | **14238** |

# 5 Conclusions

In this work, we have presented a simple yet effective Spatio-Temporal Adapter (ST-Adapter) for enabling a less studied parameter-efficient image-to-video transfer learning. Fully using commonly adopt primitive operators, ST-Adapter is particularly designed to be both lightweight and easy to implement for friendly usability and deployment. This cross-modality adaptation is a practically critical capability considering that it is dramatically challenging and more costly to build a sufficiently strong large video model in reality. Encouragingly, extensive experiments on video action recognition show that our ST-

Table 7: **Effects of kernel shape.** Kernel size is denoted as $k_T \times k_H \times k_W$ for time, height and width.

| Kernel Size | K400 | SSv2 |
|---|---|---|
| $1 \times 1 \times 1$ | 81.6 | 46.2 |
| $1 \times 3 \times 3$ | 81.4 | 46.2 |
| $3 \times 1 \times 1$ | **82.0** | **66.3** |
| $3 \times 3 \times 3$ | **82.0** | 65.6 |

Adapter can match or surpass both the full fine-tuning strategy as well as fully trained state-of-the-art video models, whilst having the benefit of (20 times less updated parameters) parameter-efficiency. Further, our method is also faster to train and consumes less computing resources with economic and environmental superiority. We believe this work is inspiring for the research of other video understanding tasks such as action localization and video summarization.

**Acknowledgement** This work is supported in part by Centre for Perceptual and Interactive Intelligence Limited, in part by the General Research Fund through the Research Grants Council of Hong Kong under Grants (Nos. 14204021, 14207319).

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
