# A   Appendix

**Implementation details of our method** All experiments are implemented in PyTorch [10]. We use the configuration listed in Tab. 1 unless otherwise specified. In general, we use much simpler data augmentation techniques compared to end-to-end fine-tuning. Hyper-parameters were briefly tuned to ensure convergence on a 20% held-out validation set.

Table 1: **Default implementation details of our method.**

| dataset and backbone | K400, ViT-B | K400, ViT-L | SSv2, ViT-B | SSv2, ViT-L |
|---|---|---|---|---|
| num. adapters per block | 1 | 1 | 2 | 1 |
| adapter bottleneck width | | 384 | | |
| convolution kernel shape | | $3 \times 1 \times 1$ $(k_T \times k_H \times k_W)$ | | |
| optimizer | | AdamW, learning rate=5e-4, weight decay=1e-2 | | |
| batch size | | 128 | | |
| training steps | 20k | 40k | 50k | 50k |
| training resize | ShortSideJitter 224 - 256 | | RandomResizedCrop | |
| training crop size | | 224 | | |
| frame sampling rate | 16 (for 8 frames per view) 8 (for 16 frames per view) 4 (for 32 frames per view) | | dynamic, evenly covering the whole video | |
| mirror | ✓ | ✓ | ✗ | ✗ |
| RandAugment [4] | ✗ | ✗ | ✓ | ✓ |
| num. testing views | 3 temporal $\times$ 1 spatial | | 1 temporal $\times$ 3 spatial | |

**Baseline implementation details** The training configuration used for all the baselines is summarized as follows:

- *Full Fine-tuning*: we largely follow the training configuration provided in their original paper, except that we train all the CLIP initialized layers with 1/100 learning rate and weight decay. We found these changes are necessary to obtain reasonable results for CLIP pretrained models; Otherwise the accuracy on Kinetics-400 is less than 50%. We found 1/100 to be the best scaling among $\{1/10, 1/100, 1/1000\}$ on Kinetics-400.

- *Partial Fine-tuning*: we finetune only the last Transformer block and the classifier layer. For the SA+TA architecture, TA is only added to the last block since the previous blocks need to be frozen in a meaningful state. We use the identical training configuration as provided in the original paper (*i.e.*, without reduction of learning rate or weight decay for any trainable weight) as we found it slightly improves accuracy for this baseline.

- Other baselines use the same training configuration as our proposed method, as stated in the *Implementation details* section in the main manuscript.

**Experiments with other foundation models** It is observed that with the same model, CLIP pre-training is superior to ImageNet21K pre-training (not surprising due to the training data scale and richness difference). However, our main objective is to propose a parameter-efficient fine-tuning alternative to the standard full fine-tuning approach particularly for image-to-video adaptation. To that end, we have validated the effectiveness and efficiency of turning an image foundation model into strong video action recognition models by tuning only a small fraction of parameters, in comparison to previous state-of-the-art alternatives.

By reporting the results on two different pre-training datasets (i.e., ImageNet21K and CLIP datasets), we would like to demonstrate that our ST-Adapter can generalize across different pre-training datasets and methods. Moreover, it can shed light on the difference between a foundation model (pre-trained with noisy web-scale raw data) and an ImageNet pre-trained model (which has been standard pre-training over the last decade).

To further support our finding, we have also experimented with the latest SWAG [11] foundation model. As seen in Table 2, our ST-Adapter with a SWAG model can achieve consistent results as

with a CLIP model: Reaching similar accuracy in the same tendency whilst outperforming the strong full fine-tuning strategy on both action datasets.

Table 2: **Experiment with different foundation models.**

| Model | Pre-train | K400 | SSv2 |
|---|---|---|---|
| ViT-B (Full Fine-tuning) | CLIP | 81.0 | 44.0 |
| ViT-B w/ ST-Adapter | CLIP | 82.0 | 67.1 |
| ViT-B (Full Fine-tuning) | SWAG | 80.1 | 45.2 |
| ViT-B w/ ST-Adapter | SWAG | 80.9 | 67.2 |

**Experiments on additional backbone architectures** we have additionally provided the results of ST-Adapters on Swin-B models in Table 3. The results of Swin space only and Swin joint attention are obtained with the training configure of [9] but using (8 frames x 3 views) sampling setting. Although they are not directly comparable with the results reported in [9] (32 frames $\times$ 12 views for K400, 32 frames $\times$ 3 views for SSv2), they are highly indicative within reasonable range. It is expected that on ImageNet-21k pretrained models our ST-Adapters underperforms full fine-tuning, especially when the locality inductive bias of Swin makes tuning on the downstream tasks easier. However, our ST-Adapter still exhibits strong temporal learning capability, matching the joint-attention Swin and outperforms space-only Swin by a large margin. Also, we observe higher data efficiency with our ST-Adapter: The Swin joint attention model on the SSv2 dataset relies on K400 pretraining (directly fine-tuning from ImageNet-21k results in slightly less than 60% accuracy). In contrast, Swin w/ ST-Adapter achieves 65.1% even when directly trained from ImageNet-21k weights. Note, we primarily aim at adapting foundation image models[2] pretrained on larger datasets (e.g., CLIP) other than ImageNet-21k.

Table 3: **Experiment with SWIN Transformers.**

| Model | K400 | SSv2 |
|---|---|---|
| Swin SpaceOnly (Full Fine-tuning) | 80.1 | 44.3 |
| Swin Join-Attention (Full Fine-tuning) | 81.5 | 65.3 |
| Swin w/ ST-Adapter | 77.1 | 65.1 |

**Inference Speed** We provide an inference speed test in Table 4. We measure the latency at batch size = 1 and throughput at batch size = 32. It is shown that our model performs slightly lower than TimeSformer space only, indicating that just a small overhead is introduced in inference speed by ST-Adapter.

Table 4: **Inference Speed.**

| Model | Total number of Params (M) | K400 | Latency (ms) | Throughput (V/s) |
|---|---|---|---|---|
| TimeSformer[1] | 121.57 | 81.7 | 28 | 69 |
| ViT-B/16 | 86.11 | 81.0 | 17 | 98 |
| ViT-B/16 w/ ST-Adapter | 93.00 | 82.0 | 19 | 90 |

**UCF-101 and HMDB-51** We verify our method on two additional smaller but also widely studied video recognition datasets, namely UCF-101 [12] and HMDB-51 [8]. For both cases, we finetune from a Kinetics-400 [3] pretrained model, with all CLIP layers fixed and ST-Adapters set to 1/10 learning rate and weight decay, and train for 500 steps with a batch size of 128. Frames are sampled with a temporal stride of 8. All other training settings are identical to that used for Kinetics-400. For testing, we use 3 spatial views and 2 temporal views, and report the 3-split mean accuracy for both datasets. We compare with methods that take only RGB frames as input (without optical flow). The

results are shown in Table 5. We observe similar top performance by our ST-Adapter in comparison to recent state-of-the-art competitors, including the latest CLIP based VideoPrompt by a large margin.

Table 5: **Comparing the state-of-the-art video recognition methods on UCF101 and HMDB51.**

| Method | Pre-train data | UCF101 | HMDB51 |
|---|---|---|---|
| STC [5] | K400 | 95.8 | 72.6 |
| ECO [16] | K400 | 93.6 | 68.4 |
| R(2+1)D-34 [13] | K400 | 96.8 | 74.5 |
| I3D [3] | ImageNet+K400 | 95.6 | 74.8 |
| S3D [14] | ImageNet+K400 | 96.8 | 75.9 |
| FASTER32 [15] | K400 | 96.9 | 75.7 |
| VideoPrompt [7] | CLIP | 93.6 | 66.4 |
| SlowOnly-8x8-R101 [6] | Kinetics+OmniSource[6] | 97.3 | 79.0 |
| ViT-B/16 w/ ST-Adapter (Ours) | CLIP+K400 | 96.4 | 77.7 |
| ViT-L/14 w/ ST-Adapter (Ours) | CLIP+K400 | 98.1 | 81.7 |
| ViT-L/14@336px w/ ST-Adapter (Ours) | CLIP+K400 | **98.3** | **82.8** |

**Visualization** We provide qualitative results about the attention map change before and after adding the ST-Adapters in Fig. 1. Videos are sampled from Something-Something-v2 dataset and the attention map of the `[CLS]` token from the last Transformer block is shown. The visualization shows that with ST-Adapters, the model attends more to action related regions (*e.g.*, hands, fore-ground objects or moving objects), while the CLIP model without adaptation tend to be distracted by the background.

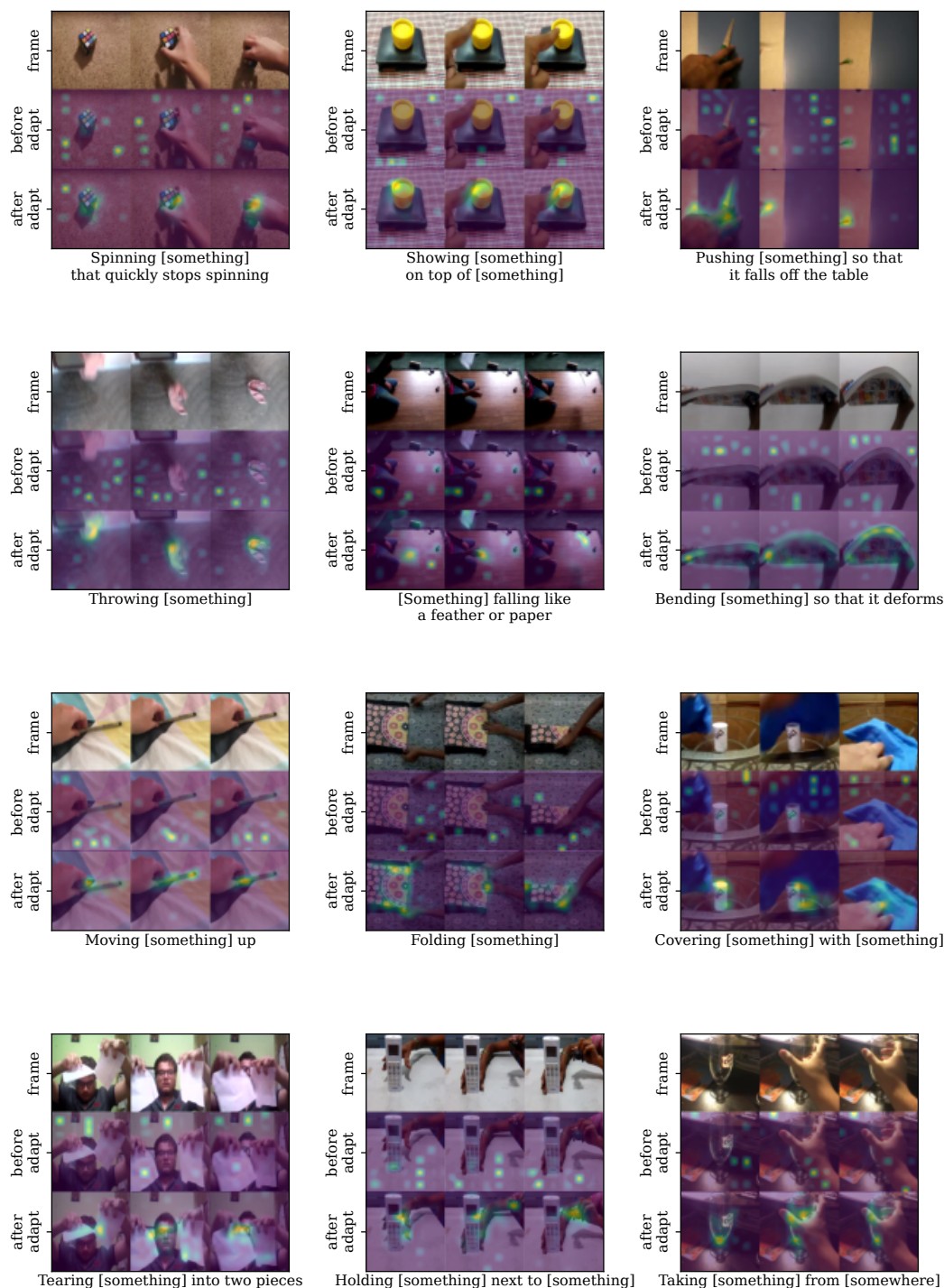

Figure 1: **Visualization of attention map before and after ST-Adaptation.**