# OpenReview forum: "ST-Adapter: Parameter-Efficient Image-to-Video Transfer Learning"
_NeurIPS.cc/2022/Conference — NeurIPS 2022 Accept_

### Official Review · Reviewer_5ouG · 2022-07-07

**Rating:** 7
**Confidence:** 4
**Soundness:** 3 good
**Presentation:** 3 good
**Contribution:** 3 good

**Summary:**

The paper proposes a parameter-efficient image-to-video transfer learning framework. In particular, the authors introduce a spatiotemporal adapter (ST-Adapter) for fine-tuning a pretrained image model to video tasks. In this setting, the parameters of the original network are frozen, and only the parameters of the  ST-Adapter are updated. The authors demonstrate that will only ~8% trainable parameters, they are able to achieve competitive results to previous fully fine-tuned approaches on various video recognition benchmarks.

**Questions:**

See the weaknesses section above.

**Limitations:**

No, the limitations were not discussed.

**Strengths And Weaknesses:**

Strengths:
+ Timely and relevant topics as video models are getting larger and larger.
+ Strong results with a small number of trainable parameters.
+ Detailed ablation studies for the most part.
+ The paper is well written and easy to follow.

Weaknesses:
- The technical contribution of the paper can be seen as somewhat limited. Adapters and other parameter-efficient modules (i.e., Lora, prompt-tuning, etc.) have been widely used in other domains such as NLP, images, etc. The authors use a relatively straightforward adaptation of these ideas (with the addition of 3D depthwise convolutions, which are commonly used for efficient video processing) to the video setting. While the technical contribution is small, given that this is the first attempt (to the best of my knowledge) to perform parameter-efficient tuning on video recognition benchmarks, I'm ok with it.
- I wish the authors validated their proposed ST-Adapter scheme with more backbones (e.g., Swin, Uniformer, Motionformer, etc.) to demonstrate its generality.
- I think it would have also been useful to see more experiments / insights into the design choices for ST-Adapter. While 3D depthwise convolution is a natural choice, the question remains is it the best choice? Have the authors experimented with any other operators (e.g., self-attention, efficient self-attention, etc.)? In my view, it would be useful to see more design-related experiments to verify that this is indeed the adapter scheme that works most effectively.
- Many of the Tables listing GFLOPs are unfair to the proposed method. Is it possible to highlight not only the GFLOPs but also the number of trained parameters? The proposed approach is trained using significantly fewer parameters than the remaining baselines that are fully finetuned.

---

> ### Author Response · Authors · 2022-08-02
> **Reply to Reviewer  5ouG**
>
> We thank the reviewer for all the constructive comments. We address all the concerns below:
>
> **Q1. Experiments on additional backbone architectures**
>
> A1. Please see our response to Reviewer HAPR Q2 and Q3 for a detailed discussion.
>
> **Q2. Structural ablation studies on the ST-Adapter module**
>
> A2. Thanks. As suggested, we have now provided the results by replacing the convolutions with self-attention modules for temporal modeling. Temporal positional embeddings are also added before each self-attention module. To keep the number of parameters comparable, we use 6 heads each with channel size of 32. We report the top-1 results.
>
> |                  | K400     | SSv2     |
> | ---------------- | -------- | -------- |
> | DW-Conv 3x1x1    | **82.0** | **66.3** |
> | Time-only Attn.  | **82.0** | 65.9     |
> | Space-time Attn. | 81.8     | 63.9     |
>
> As shown in the table above, despite being simple and efficient, the depthwise convolutions offer the best accuracy. We conjecture that more complex temporal learning design may need more training data to be effective.
>
> **Q3. Comparing number of trained parameters with state-of-the-art methods**
>
> A3. Thanks. We will include. Given that most existing methods take a full fine-tuning scheme, our method is significantly advantageous in terms of the number of trained parameters.

---

### Official Review · Reviewer_HAPR · 2022-07-11

**Rating:** 5
**Confidence:** 5
**Soundness:** 3 good
**Presentation:** 3 good
**Contribution:** 3 good

**Summary:**

Authors propose an extension to the Adapter [24] framework to build a space-time adapter, which converts a pretrained image model into a video recognition model by finetuning very few parameters on the video task. The proposed adapter is very simple, based upon depthwise 3D convolution, added to each Transformer block. Experiments are shown on Kinetics and SSv2, where the proposed method obtains strong results, often comparable to fully finetuning the network (less so on SSv2, given its temporally challenging recognition task). Moreover, the resulting model is more data and training efficient.

**Questions:**

Please reply to the weaknesses above, especially 1 and 2.

**Strengths And Weaknesses:**

## Strengths
1. [originality] While the method is a simple extension of "Adapter" [26]: The proposed method is very simple to implement and builds upon a well known Adapter architecture. Nevertheless, it obtains strong results and to my knowledge these results are not well known in the community.
2. [quality] **Comprehensive ablations:** Authors provide comprehensive set of interesting ablations. This includes apples to apples comparisons with reasonable baselines (Table 1), as well as properties of the adapter in Fig 2 and Tab 4, which show STAdapter makes the model training and data efficient.
3. [clarity] **The paper is generally very well written**, easy to follow, with useful and easy to read figures.

## Weaknesses
1. [quality] **Runtime savings:** While authors show impressive results in reducing parameters being finetuned yet obtaining near SOTA performance, it is unclear what the real-world advantages of such a method is. Does the reduced parameters being trained lead to speed up in training time? If so, by how much? Does that make the training possible on GPUs with less memory? If the proposed approach makes SOTA video models accessible to researchers with constrained resources, the impact of the paper will be much more.
2. [quality] **Other backbones:** The results are largely limited to ViT based models. It's not clear if the approach can also work with other popular architectures, such as MViT and Swin? Swin for instance does come with large scale pretrained models, which would be interesting to adapt to videos.
3. [significance] **Results without CLIP are not that strong**: Authors show impressive results, including 85.6 on Kinetics, while finetuning very few parameters and operating on very few frames from CLIP pretrained model. However, the results with ImageNet-21K are not as strong (although they do obtain improvements over other approaches for parameter efficient finetuning). It's not clear why this is the case. Perhaps more results with other "foundation models" (such as https://github.com/facebookresearch/SWAG) could help decipher if this approach is best applicable to vision-text models or does it work more generally.
4. [clarity] **SOTA results missing** some higher performing methods like "Multiview Transformers for Video Recognition" (CVPR'22), "Omnivore: A Single Model for Many Visual Modalities" (CVPR'22) etc. While not directly comparable since they use different architectures and pre-training, they obtain better performance in Table 2 and 3 and should be reported when comparing to SOTA.
5. [significance] **More datasets:** Since the proposed method is fairly simple and general, it would be nice to evaluate it more broadly for other video understanding tasks, such as egocentric action classification (eg EPIC, Ego4D), long video classification (eg HowTo100M). Such experiments would further throw light on the advantages and limitations of the proposed adapter.

## Minor
- L251: "arts" -> "art"
- L256 "stat" -> "state"

---

> ### Author Response · Authors · 2022-08-02
> **Reply to Reviewer HAPR**
>
> We thank the reviewer for all the constructive comments. We address all the concerns below:
>
> **Q1. Runtime saving of ST-Adapter training**
>
> A1. Thanks, great suggestion. We report the total training GPU-hours and peak memory usage for three models: ST-Adapter, TimeSformer space only, and TimeSformer divided space time (ViT-B/16, 8 input frames, batch size 16) in the table below:
>
> | Methods| Training GPU-hours (K400)  | Peak mem (M) |
> | ---------------------- | -------------- | ------------ |
> | ST-Adapter                     | **23**     | **14238**    |
> | TimeSformer space only         | 40 (+74%)   | 17275 (+21%) |
> | TimeSformer divided space time | 60 (+161%)  | 21694 (+52%) |
>
> Except the training time in our default settings tested as above, we can also apply ST-Adapter to deeper blocks and largely maintain the accuracy (see Table 4b), which may further reduce the backward propagation time. ST-Adapter may also scale to a larger number of GPUs as less parameters need to be synchronized among workers.
>
> **Q2. Experiments on additional backbone architectures**
>
> A2. Thanks. As suggested, we have additionally provided the results of ST-Adapters on Swin-B models in the table below. The results of Swin space only and Swin joint attention are obtained with the training config of [a] but using (8 frames x 3 views) sampling setting. Although they are not directly comparable with the results reported in [a] (32 frames x 12 views for K400, 32 frames x 3 views for SSv2), they are highly indicative within reasonable range.
>
> | Model | K400 | SSv2 |
> | ------ | ------ | ------ |
> | Swin space-only | 80.1 | 44.3 |
> | Swin joint-attention (window=4x7x7) | 81.5 | 65.3 |
> | Swin w/ ST-Adapter | 77.1 | 65.1 |
>
> It is expected that on ImageNet-21k pretrained models our ST-Adapters underperforms full fine-tuning, especially when the locality inductive bias of Swin makes tuning on the downstream tasks easier. However, as shown in the table above, our ST-Adapter still exhibits strong temporal learning capability, matching the joint-attention Swin and outperforms space-only Swin by a large margin.
> Also, we observe higher data efficiency with our ST-Adapter: The Swin joint attention model on the SSv2 dataset relies on K400 pretraining (directly finetuning from ImageNet-21k results in slightly less than 60% accuracy). In contrast, Swin w/ ST-Adapter achieves 65.1% even when *directly trained from ImageNet-21k weights*.
> Again, we stress that we primarily aim at adapting foundation image models [8] pretrained on larger datasets (e.g., CLIP) other than ImageNet-22k.
>
> [a] Liu, Ze, et al. "Video swin transformer." CVPR 2022.
>
> **Q3. Experiments with other ‘foundation models’**
>
> A3. We are sorry that we were not able to finish the experiments within the first week. We will continue to update the results of these experiments during the discussion period. Regarding the difference between CLIP pretrain and ImageNet pretrain, we provided more detailed dicussion in the reply for Q2 of Reviewer ZMxj.
>
> **Q4. Additional SOTA results**
>
> A4. Thanks. We will include these works suggested.
>
> ​​**Q5. Additional Datasets**
>
> A5. Thanks. As suggested, we have now provided the experiments on Epic-Kitchens-100 in the following table.
>
> | Model                | Pretrain   | #Frames | Verb     | Noun     |
> |----------------------|------------|---------|----------|----------|
> | ViViT-L[3]           | IN21K+K400 | 16x3x10 | 66.4     | **56.8** |
> | MFormer-B[51]        | IN21K+K400 | 16x3x10 | 66.7     | 56.5     |
> | X-ViT(8x)[10]        | IN21K+K400 | 8x3x1   | 66.7     | 53.3     |
> | ViT-B w/o ST-Adapter | CLIP       | 8x3x1   | 54.8     | 50.4     |
> | ViT-B w/ ST-Adapter  | CLIP       | 8x3x1   | **67.6** | 55.0     |
>
> It is evident that our ST-Adapter consistently brings a big margin. Also, we found that without our ST-Adapter, it is much more difficult to directly adapt CLIP pretrained ViT on the domain of egocentric video with high sensitivity to the hyperparameter setting. ST-Adapter eases the training process. It is worthy to note that, all current transformer based approaches need to be pre-trained first on image dataset and then fine-tuned on Kinetics dataset before fine-tuned with egocentric videos, in contrast our ST-Adapter is directly applied to a image model and trained with target egocentric data only.

---

> > ### Comment · Reviewer_HAPR · 2022-08-08
> > **Rebuttal response**
> >
> > I appreciate authors' response to my concerns. It is good to see the proposed method does make the train time faster, and works on egocentric videos without needing intermediate Kinetics pretraining (although I'd argue that is not that surprising, given CLIP probably has seen many Kinetics-like video frames). I intend to keep my original rating.

---

> > > ### Author Response · Authors · 2022-08-08
> > > **Thanks to the reviewer for giving further comments!**
> > >
> > > Thanks to the reviewer for giving further comments! We provide experimental results with ViT-B pre-trained with SWAG as suggested.
> > >
> > > | Model | Pretrain|K400 | SSv2 |
> > > | - | - | - | - |
> > > | ViT-B full fine-tuning | CLIP | 81.0     | 44.0     |
> > > | ViT-B w/ ST-Adapter  | CLIP  | **82.0** | 66.3 |
> > > | ViT-B full fine-tuning | SWAG | 80.1 | 45.2 |
> > > | ViT-B w/ ST-Adapter | SWAG | 80.9 | **67.2** |
> > >
> > > As seen in the Table above, our ST-Adapter with SWAG achieves consistent results as with CLIP: Reaching similar accuracy in the same tendency whilst outperforming the strong full fine-tuning strategy on both action datasets. This suggests that our method can generalize well over different pretraining datasets and methods.
> > >
> > > We hope that these additional results can address the concerns raised by the reviewer and are helpful in making the final recommendation.
> > > Please let us know if more comments exist and we are happy to respond as quickly as possible.

---

### Official Review · Reviewer_Y5Pa · 2022-07-11

**Rating:** 5
**Confidence:** 5
**Soundness:** 2 fair
**Presentation:** 3 good
**Contribution:** 2 fair

**Summary:**

The goal of this paper is to perform parameter-efficient learning on image-to-video adaptation tasks. Specifically, the authors propose a Spatio-Temporal Adapter, which applies an additional depth-wise 3D convolution in adapters, to incorporate temporal and spatial information with only a limited amount of parameters in the model fine-tuned. In the experiment, the authors showed that, compared with full fine-tuning and other fine-tuning techniques, the proposed ST-adapter can achieve better results in Kinetics-400 and Something-Something-v2 datasets.

**Questions:**

The following questions are mainly based on the concerns listed in the Weaknesses part. Here are the questions:

- I have some concerns regarding the experimental setup (i.e., doing early-stoping on the validation set and only presenting validation set results for parameter-efficient methods). Could the authors provide the test set results (or doing early-stopping on the held-out set) or provide arguments to alleviate this concern?

- Could the authors present a column to clearly show the number of total parameters across different methods? How does the extra parameter affect the inference time compared to the method without the extra parameters?

- To have a fairer comparison to SoTA, could the authors provide the results under the same setups, such as results of (1) ViT-L pretrained on IM-21k and applying ST-Adapter and (2) ViT-L pretrained on Clip and without ST-adapter?


- Notation is confusing. In line 120, the authors mention $N=H \times W / P^2$ which is the number of patches in a single frame without temporal information. However, in line 173, the authors mention the feature representations will be first ***reshaped*** from $X' \in R^{N \times d}$ to $R^{T \times H \times W \times d}$. Why does $N$ contain the temporal information? There seems to be notation confusion or inaccurate description.



**Limitations:**

I do not see the authors including any discussion on limitations or negative societal impact. The potential limitations of the paper are the inference time of using extra parameters and generalization to other video tasks.

**Strengths And Weaknesses:**

Strengths:
- The motivation is reasonable. Since the video pre-training data or pretrained models are limited or hard to obtain, it is ideal to use pretrained models/data from other modalities and adapt to video domains.

- The paper is easy to understand, and the presentation is clear and good.

Weaknesses:
- Since the major difference between the prior work [1] and the proposed method is depth-wise 3D convolutional layers in the hidden state, it is important to compare the prior work and show that such a design makes a difference in the performance.

- A major weakness of this paper is the experimental setup. As mentioned in the prior empirical study [2] on parameter-efficient methods, a common pitfall or mistake of the existing parameter-efficient works is only presenting validation set results while doing early-stoping on the validation set. This makes the reported results of the parameter-efficient methods biased on the validation set and higher than the full fine-tuning. A more reasonable experimental setup is to either report test set results or split the training set and perform early-stoping on the held-out set. It is unclear whether this paper follows the proper experimental setup.

- In Table 1, only showing the number of fine-tuned parameters is a bit misleading, as some methods including ST-adapter introduces **extra** parameters which makes the comparison unfair due to the different amounts of total parameters in the models across different methods.

- Comparison to the state-of-the-art methods is confusing. Since the major focus is the effectiveness of the ST-adapter, a fair comparison should be, under the same experiment setups (e.g., same pretrained datasets), how the ST-adapter improves against the prior works. For example, the authors should present the results of (1) ViT-L pretrained on IM-21k and apply ST-Adapter and (2) ViT-L pretrained on Clip and without ST-adapter. In this way, the reader can understand the improvement gains solely provided by the proposed ST-adapter.

[1] "Parameter-efficient transfer learning for nlp", Houlsby et al., ICML '19

[2] "Revisiting Parameter-Efficient Tuning: Are We Really There Yet?", Chen et al.

---

> ### Author Response · Authors · 2022-08-02
> **Reply to Reviewer Y5Pa**
>
> We thank the reviewer for all the constructive comments. We address all the concerns below:
>
> **Q1.Comparison with original Adapter in [1]**
>
> A1. Thanks for the suggestion. The comparison is given in the table below.
>
> | Methods     | Architecture | Fine-tuned Params(M) | K400 | SSv2 |
> |-------------|--------------|----------------------|------|------|
> | Adapter[1]  | SA           | 6.77                 | 81.6 | 46.2 |
> | ST-Adapter  | SA           | 7.20                 | 82.0 | 66.3 |
>
> We can see that our method outperforms Adapter[1] on both datasets, particularly on SSv2 with a need for demanding temporal reasoning.
>
> **Q2.It is unclear whether this paper follows the proper experimental setup.**
>
> A2. Overall, we follow the existing works in experiments for fair comparison. To avoid overfit to the standard validation set, we use the following experiment protocol: we first split the original training set into two subsets: 80% for training and 20% for hyper-parameter tuning. Once the optimal parameters are obtained, we perform the final training on the full training set and test on the standard validation set. This ensures that our experiment does not break the basic machine learning principles.
>
> We would also like to stress that the two datasets we used (K400 and SSv2) are large (with about 240k/20k and 160k/25k training/validation samples respectively), and early stop is generally ineffective due to the low variance of accuracy among multiple runs, unlike the case in [a]. For instance, on Kinetics-400, repeatedly running the same config typically results in an accuracy difference of less than 0.1%.
>
> [a] Chen et al.  "Revisiting Parameter-Efficient Tuning: Are We Really There Yet?" arXiv 2022.
>
>
> **Q3.  The difference in number of model parameters makes the comparison in Table 1. unfair.**
>
> A3. The number of parameters are generally different across methods. Note that for all efficient tuning approaches (all fine-tuning methods in Table 1 except full fine-tuning) have the same frozen backbone. Therefore, we consider that removing such common part whilst focusing on only the fine-tuned parameters can better contrast the difference between different transfer learning strategies.
>
> Please also note that, in terms of the total number of parameters, our proposed ST-Adapter only adds 8.3% of parameters in addition to the original ones of the backbone which is the same for all approaches in Table 1.
>
> Please note there is not a strong correlation between the number of parameters and inference speed. Nevertheless, we have now provided a inference speed test. We measure the Latency at batch_size = 1 and Throughput at batch_size = 32. The table below shows that our model performs slightly lower than TimeSformer space only, indicating that just a small overhead is introduced in inference speed by ST-Adapter.
>
> | Model | Total number of params (M) | Acc. K400 | Latency (ms) | Throughput V/s |
> | --- | --- | --- | --- | --- |
> | TimeSformer space only | 86.11 | 81.0 | 17 | 98 |
> | TimeSformer divide space time | 121.57 | 81.7 | 28 | 69 |
> | ViT-B/16 w/ ST-Adapter | 93.00 | 82.0 | 19 | 90 |
>
>
>
> **Q4. How the ST-adapter improves against the prior works.**
>
> A4. Sorry for the confusion. We have already provided a comparison between ViT-B w/o ST-adapter (Full Fine-tuning SA ) and ViT-B w/ adapter (ST-Adapter(ours) SA) in Table 1 (main paper). We will refine Table 2 and Table 3 with revised notation as suggested by the reviewer. In the following table we provide the comparison between (1) ViT-B pretrained on IN-21k and CLIP with ST-adapter and (2) ViT-B pretrained on IN-21k and CLIP without ST-adapter. The results suggest that our method is generally effective.
>
> | Model                | Pretrain | #Frames | K400     | SSv2     |
> |----------------------|----------|---------|----------|----------|
> | ViT-B w/o ST-Adapter | IN21K    | 8x3x1   | 76.9     | 40.0     |
> | ViT-B w/ ST-Adapter  | IN21K    | 8x3x1   | 76.5     | 62.8     |
> | ViT-B w/o ST-Adapter | CLIP     | 8x3x1   | 81.0     | 44.0     |
> | ViT-B w/ ST-Adapter  | CLIP     | 8x3x1   | **82.0** | **66.3** |
>
> Given the short rebuttal period we do not have enough resources to conduct experiments on ViT-L, since it needs significantly more computation and time. However, we will include this comparison in the final version.
>
> **Q5. Notation typo in line 173.**
>
> A5. Thanks for pointing it out and sorry for the confusion. The correct notation should be:
>
> $\mathbf{X'} \in \mathbb{R}^{T \times N \times d}$ to  $\mathbf{X''} \in \mathbb{R}^{T \times h \times  w \times d}$  where $ N = h \times w$.

---

> > ### Comment · Reviewer_Y5Pa · 2022-08-08
> > **Thanks authors for providing the experiments**
> >
> > Thanks authors for providing the experiments. I understood that the authors have shown adding ST-Adapter can improve vanilla ViT-B in Table 1, but it would be clearer to distinguish the improvement gain from pretrained dataset and proposed ST-Adapter.
> >
> > Overall, most of my concerns have been addressed, while I would suggest the authors include ViT-L experiment in Table 2 and Table 3. I would increase my rating to borderline accept.

---

> > > ### Author Response · Authors · 2022-08-08
> > > **Thanks to the reviewer for giving further comments and lifting the score!**
> > >
> > > Thanks to the reviewer for giving further comments and lifting the score!  As promised in the response, we will add the results using ViT-L as the foundation model w/ and w/o our ST-Adapter (as this experiment is time consuming).
> > >
> > > We agree that different pre-train datasets would contribute differently to the final performance. It is observed that with the same model, CLIP pre-training is superior to ImageNet21K pre-training (not surprising due to the training data scale and richness difference). However, we want to clarify that our main objective is to propose a parameter-efficient fine-tuning alternative to the standard full fine-tuning approach particularly for image-to-video adaptation. To that end, we have validated the effectiveness and efficiency of turning an image foundation model into strong video action recognition models by tuning only a small fraction of parameters, in comparison to previous state-of-the-art alternatives.
> > >
> > > By reporting the results on two different pre-training datasets (i.e. ImageNet21K and CLIP datasets), we would like to demonstrate that our ST-Adapter can generalize across different pre-training datasets and methods. We have validated this point in our paper. Moreover, it can shed light on the difference between a foundation model (pre-trained with noisy web-scale raw data) and an ImageNet pre-trained model (which has been standard pre-training over the last decade).
> > >
> > > To further support our finding, as suggested by reviewer HAPR, we have also experimented with the latest SWAG foundation model. As seen in the Table below, our ST-Adapter with a SWAG model can achieve consistent results as with a CLIP model: Reaching similar accuracy in the same tendency whilst outperforming the strong full fine-tuning strategy on both action datasets.
> > >
> > > | Model | Pretrain|K400 | SSv2 |
> > > | - | - | - | - |
> > > | ViT-B w/o ST-Adapter | SWAG | 80.1 | 45.2 |
> > > | ViT-B w/ ST-Adapter | SWAG | 80.9 | 67.2 |
> > >
> > > We hope our clarification and the new experiments would address the remaining concerns. Please let us know if more comments exist and we are happy to respond as quickly as possible.

---

> > > > ### Author Response · Authors · 2022-08-09
> > > > **Additional results for ViT-L w/o ST-Adapter**
> > > >
> > > > Dear reviewer,
> > > >
> > > > As suggested, we are now providing more results of ViT-L w/o ST-Adapter in the following table:
> > > >
> > > > | Model                                 | Pretrain | K400 | SSv2 |
> > > > |---------------------------------------|----------|------|------|
> > > > | ViT-L w/o ST-Adapter (Full Fine-tune) | CLIP     | 85.8 | 48.7 |
> > > > | ViT-L w/ ST-Adapter                   | CLIP     | 86.7 | 70.0 |
> > > >
> > > > Given the short rebuttal period we do not have enough resources to conduct experiments on ViT-L pre-trained on ImageNet-21K. However, we will include this comparison in the final version. Thanks for your understanding!

---

### Official Review · Reviewer_ZMxj · 2022-07-11

**Rating:** 6
**Confidence:** 3
**Soundness:** 3 good
**Presentation:** 3 good
**Contribution:** 3 good

**Summary:**

This paper proposes a new Spatio-Temporal Adapter (ST-Adapter) for parameter-efficient fine-tuning on video tasks. With a much smaller trainable parameter, ST-Adapter can match or even outperform the strong full fine-tuning strategy on K400 and SSv2 datasets.


**Questions:**

See Strengths & Weakness.

**Limitations:**

No.

**Strengths And Weaknesses:**

Pros:
1. It's a good attempt to introduce an Adapter for Spatio-Temporal modeling using a 2D pretrained model.
2. The proposed ST-Adapter are effective. It can match or even outperform the strong full fine-tuning strategy with a much smaller trainable parameter.
3. SOTA results on K400 and SSv2 using a CLIP pre-trained model.

Cons:
1. Lack two important comparisons. The new added ST-Adapter has a  quite different structure from common practice for extending a 2D backbone to Spatio-temporal, e.g, an additional temporal self-attention in Figure1. So it would be better to disentangle the effect of the new structure (e.g., DW 3D-CONV which may have a complementary impact on a full Transformer architecture) and learning strategy (only tuning some new added modules, which is also called Adapter). So, exp1 [finetuning the (b) of Figure 1] and exp2 [using (a) as an Adapter of Figure 1] would be valuable.
2. CLIP pre-trained model and ImageNet-21K pretrained model performs very differently. For example, Partial Fine-tuning using CLIP pre-trained model on K400 only performs a little worse than Full Fine-tuning (80.1 vs 81.0) while this gap is quite large when using ImageNet-21k pre-trained model (61.7 vs 76.9). It seems that the CLIP learns many video concepts which may narrow the gap between pre-training and video finetuning, especially for K400 which is largely biased to spatial appearance. So it may be more suitable to conduct an ablation study using ImageNet-21K pretrained model or using SSv2 dataset.
3. Similar to 2, it would be helpful to add the ImageNet-22K pre-trained results in Table 2 and Table 3

---

> ### Author Response · Authors · 2022-08-02
> **Reply to Reviewer ZMxj**
>
> We thank the reviewer for all the constructive comments. We address all the concerns below:
>
> **Q1. Lack two important comparisons: (1) Full finetuining with ST-Adapter and (2) Finetuning only the temporal attention modules of Fig. 1a.**
>
> A1.  Thanks for the suggestions. The two comparisons can be found in the table below.
> | Methods                       | Architecture | Fine-tuned Params(M) | K400 | SSv2 |
> |-------------------------------|--------------|----------------------|------|------|
> | Temporal Fine-tuning          | SA+TA        | 35.8                 | 81.3 | 59.4 |
> | ST-Adapter (Full Fine-tuning) | SA           | 93.31                | 82.3 | 67.3 |
> | ST-Adapter                    | SA           | 7.20                 | 82.0 | 66.3 |
>
> Method (1) Is denoted as ST-Adapter (Full Fine-tuning) – All parameters are finetuned including those of the original backbone and additional parameters introduced by our ST-Adapter. Method (2) is denoted as Temporal Fine-tuning – Only tuning the temporal attention modules in the SA+TA architecture. We can observe from the table that while ST-Adapter (Full Fine-tuning) performs the best at a higher training cost, it is not always feasible given the increasing size of foundation models. Our model is superior to Temporal Fine-tuning on both K400 and SSv2 while training much less parameters.
>
>
> **Q2. CLIP pre-trained model and ImageNet-21K pretrained model performs very differently.**
>
> A2. It is true that the CLIP pretrained models and ImageNet-21K models perform very differently. This is mainly because the (vision+language) training data of CLIP is significantly richer in terms of knowledge and diversity as compared to image only ImageNet-21K. In this sense, the CLIP model is considered as a more proper foundation model [8]. This means ImageNet-21K is limited in the availability of knowledge to be transferred. This explains well their performance difference in the downstream task transfer. Given this, it is reasonable that we selected the CLIP model for ablation study.
>
> **Q3. it would be helpful to add the ImageNet-22K pre-trained results in Table 2 and Table 3**
>
> A3. Thanks for the suggestion, we will include ImageNet-21K pretraining based results in Table 2 and Table 3 (currently only in Table 1).

---

> > ### Comment · Reviewer_ZMxj · 2022-08-09
> > **Response**
> >
> > Thanks authors for preparing the rebuttal. My concerns are well addressed and thus I keep my initial rating.

---

### Author Response · Authors · 2022-08-09
**Thanks all reviewers for the constructive feedbacks!**

We thank all reviewers for constructive feedback. We appreciate the positive comments that (1) Our results are strong despite fine-tuning only a small fraction of parameters (ZMxj, Y5Pa, HAPR, 5ouG); (2) Comprehensive ablations (HARP, 5ouG); (3) The paper is well written and easy to follow (Y5Pa, HAPR, 5ouG). We have carefully revised the manuscript to be of significantly higher quality according to the suggestions. All changes are in blue for easy tracking and further review.

---

### Meta-Review · Area_Chair_sKkX · 2022-08-26

**Recommendation:** Accept
**Confidence:** Certain

**Metareview:**

This paper proposes a new Spatio-temporal adapter for parameter-efficient fine-tuning per video task transferred from a pretrained image model. After the discussion phase, the requested comparisons on full finetuning, finetuning only the temporal attention modules and more backbones are added and the reviewers are satisfied with the rebuttal. Given all the positive scores by reviewers, the meta-reviewers recommend accepting this paper.

**Award:**

No

---

### Decision · Program_Chairs · 2022-09-14

Accept